# Map-Based Cloning, Phylogenetic, and Microsynteny Analyses of *ZmMs20* Gene Regulating Male Fertility in Maize

**DOI:** 10.3390/ijms20061411

**Published:** 2019-03-20

**Authors:** Yanbo Wang, Dongcheng Liu, Youhui Tian, Suowei Wu, Xueli An, Zhenying Dong, Simiao Zhang, Jianxi Bao, Ziwen Li, Jinping Li, Xiangyuan Wan

**Affiliations:** 1Biology and Agriculture Research Center, University of Science and Technology Beijing, Beijing 100024, China; b20180389@xs.ustb.edu.cn (Y.W.); dongchengliu@ustb.edu.cn (D.L.); tianyouhui@ustb.edu.cn (Y.T.); suoweiwu@ustb.edu.cn (S.W.); xuelian@ustb.edu.cn (X.A.); zydong@ustb.edu.cn (Z.D.); zhangsimiao11@126.com (S.Z.); bjx1232003@126.com (J.B.); ziwen.li@163.com (Z.L.); 2Beijing Engineering Laboratory of Main Crop Bio-Tech Breeding, Beijing International Science and Technology Cooperation Base of Bio-Tech Breeding, Beijing Solidwill Sci-Tech Co. Ltd., Beijing 100192, China; lijinping@sjlhtech.com

**Keywords:** genic male sterility (GMS), anther development, glucose methanol choline (GMC) oxidoreductase, hybrid breeding, *Zea may*

## Abstract

Genic male sterility (GMS) mutant is a useful germplasm resource for both theory research and production practice. The identification and characterization of GMS genes, and assessment of male-sterility stability of GMS mutant under different genetic backgrounds in *Zea may* (maize) have (1) deepened our understanding of the molecular mechanisms controlling anther and pollen development, and (2) enabled the development and efficient use of many biotechnology-based male-sterility (BMS) systems for hybrid breeding. Here, we reported a complete GMS mutant (*ms20*), which displays abnormal anther cuticle and pollen development. Its fertility restorer gene *ZmMs20* was found to be a new allele of *IPE1* encoding a glucose methanol choline (GMC) oxidoreductase involved in lipid metabolism in anther. Phylogenetic and microsynteny analyses showed that ZmMs20 was conserved among gramineous species, which provide clues for creating GMS materials in other crops. Additionally, among the 17 maize cloned GMS genes, *ZmMs20* was found to be similar to the expression patterns of *Ms7*, *Ms26*, *Ms6021*, *APV1*, and *IG1* genes, which will give some clues for deciphering their functional relationships in regulating male fertility. Finally, two functional markers of *ZmMs20/ms20* were developed and tested for creating maize *ms20* male-sterility lines in 353 genetic backgrounds, and then an artificial maintainer line of *ms20* GMS mutation was created by using *ZmMs20* gene, *ms20* mutant, and BMS system. This work will promote our understanding of functional mechanisms of male fertility and facilitate molecular breeding of *ms20* male-sterility lines for hybrid seed production in maize.

## 1. Introduction

Male sterility is a common phenomenon in higher plants and widely used in crop hybrid seed production. According to the inheritance characteristic, it can be divided into cytoplasmic male sterility (CMS) and genic male sterility (GMS). Generally, CMS is caused by a toxic protein in the cytoplasm, and the fertility can be rescued by restorer genes in the nucleus. GMS usually results from the loss-function of genic genes, mainly participating in anther and pollen development. Anther locules are composed of center-localized meiocyte cells, which are surrounded by four somatic cell layers. From stamen primordia initiation to mature pollen formation, thousands of genes likely take part in this complicated process. So far, hundreds of GMS mutants have been found, while only 17 GMS genes have been cloned in maize. Among them, nine genes were reported to participate in the ontogeny of anther cuticle and pollen exine [1].

The anther cuticle locates outside of the epidermis. It acts as a barrier to transpiration, pathogens, and other stresses [2]. The anther cuticle contains cutin and wax. Cutin is mainly composed of polyesters derived from hydroxylated and epoxy C16 and C18 fatty acids produces [3], and cuticle wax is a mixture of very-long-chain fatty acid, alkanes, alkenes, and fatty alcohols [4]. To date, some GMS genes participating in anther cuticle development have been isolated in model plants, such as *Arabidopsis thaliana*, *Oryza sativa* (rice), and maize. For instance, *AtMs2*, *OsDPW*, and *ZmMs6021* encode plastid localized reductases catalyzing the reduction of fatty acyl-ACP to corresponding fatty alcohols [5,6,7]. *AtCYP703A2*, *OsCYP703A3*, and *ZmAPV1* function as lauric acid hydroxylase [8,9,10]. *AtCYP704B1*, *OsCYP704B2*, and *ZmMs26* catalyze the ω-hydroxylation of fatty acid [11,12,13]. *OsGPAT3* and *ZmMs33* encode glycerol-3-phosphate acyltransferase with unknown substrates [14,15,16]. *ZmMs30* encodes a Gly-Asp-Ser-Leu (GDSL)-motif lipase with diverged catalytic residues and prefers substrates with short carbon chains [17]. The loss-function of any of the genes mentioned above leads to arrested anther cuticle development. In addition, transcription factors, such as *AtMs1*, *OsPTC1*, and *ZmMs7*, were also reported to regulate anther cuticle development [18,19,20].

Glucose-methanol-choline (GMC) oxidoreductase family catalyze the oxidation of CH-OH to the corresponding aldehyde. The GMC oxidoreductase family includes diverse subclades, such as oxidoreductases, dehydrogenases, lyases, and oxidases [21]. Generally, these members have a wide variety of substrates. For instance, the oxidases can be divided into glucose oxidase, choline oxidase, cholesterol oxidase, cellobiose dehydrogenase, aryl-alcohol oxidase, and pyridoxine 4-oxidase according to the substrates [22]. An algae-specific GMC member can catalyze the decarboxylation of free fatty acids to n-alkanes or n-alkenes in response to blue light [23]. In plants, *Arabidopsis ACE*/*HTH* is the first reported GMC gene, and it oxidizes long-chain ω-hydroxy fatty acids produced in the ω-oxidation pathway of cytochrome P450 fatty acid ω-hydroxylases [24]. Its transcripts are detectable in both floral organs and roots; however, *ace*/*hth* only shows fusions of floral organs and genetic background-dependent male sterility due to the disruption of the membrane structure [24]. Rice *mini1*, an ortholog of *ace*/*hth*, is more severely defective, and exhibits a smaller architecture, shortened leaves and sheaths, and withered leaf blade tips [25,26]. *HTH1* and *OsNP1*, two other homologous GMC genes specifically expressed in the anther of rice, have been recently reported to participate in anther cuticle development. *HTH1* is highly expressed in the anther epidermis and tapetum, and *OsNP1* is expressed in the anther tapetum. Both *hth1* and *np1* display a glossy anther surface and aborted pollen grains [27,28,29]. In maize, *IPE1* exhibits similar characteristics to its ortholog *OsNP1* [30].

In this study, we found that *ms20*, a complete male-sterility mutant, displayed an unextruded smaller anther, glossy cuticles, and abnormal pollen grains. *ZmMs20* was found to be a new allele of *IPE1* and encode a GMC oxidoreductase, which is a small gene family including 8, 7, and 6 members in *Arabidopsis*, rice, and maize, respectively. To explore the potential functions of GMC genes underlying anther development, phylogenetic and expression pattern analyses were performed. The analysis results showed that *ZmMs20* and its orthologs underwent different evolution pathways between monocots and dicots, while they were greatly conserved during the evolution of gramineous species. Additionally, the expression patterns of 17 cloned GMS genes in maize were investigated here. Most of these genes were expressed in anthers with different expression peaks at stages 7, 8, or 9. *ZmMs20* was found to be expressed after initiation of meiosis with high expression at stages 8b and 9, similar to the expression patterns of *Ms7*, *Ms26, Ms6021, APV1*, and *IG1* genes [7,10,13,20,31]. Finally, two functional markers were developed to create maize *ms20* male-sterility lines in different genetic backgrounds by marker-assisted selection (MAS), which could facilitate molecular breeding of *ms20* male-sterility lines for hybrid breeding and seed production in maize.

## 2. Results

### 2.1. Genetic and Phenotypic Analyses of ms20 Mutant

Maize male-sterility mutant *ms20* was originally obtained from the Maize Genetics Operation Stock Center (http://maizecoop.cropsci.uiuc.edu). When the mutant plants were crossed with maize inbred line Chang7-2, all the F_1_ progenies were fertile. The segregation of fertile to sterile F_2_ individuals fitted an approximate ratio of 3:1 (Table 1), which uncovered the recessive monofactorial inheritance characteristic of *ms20* mutant. Compared with wild type (WT), no obvious difference was observed in *ms20* during vegetative growth. However, *ms20* anthers could not be exerted out of glumes (Figure 1A1,A2), became smaller and wilted (Figure 1A3,A4), and did not generate mature pollen grains (Figure 1A5,A6). The seed setting of *ms20* was normal when pollinated with WT pollens, indicating that the female fertility of *ms20* was unaffected.

### 2.2. Defective Anther Cuticle and Abortion Pollen Grain in ms20 Mutant

To explore defects of *ms20* anther and pollen, scanning electron microscope (SEM) analysis was performed at maize anthesis stage. Consistent with the above morphological results, *ms20* anther was smaller and wilted (Figure 1B2) compared with that of WT (Figure 1B1). In addition, both the outer surface (Figure 1B4) and inner surface (Figure 1B6) of *ms20* anther were glossy and smooth, and no typical Spaghetti-like anther cutin pattern (Figure 1B3) and Ubisch bodies (Figure 1B5) were observed in *ms20* anther. In WT, pollen grains with round shape (Figure 1A5,B7,C1) were generated for double fertilization, while only severely wizened pollen grains were retained and pasted on the inner wall of the anther locule (Figure 1A6,B8,C2).

### 2.3. Isolation of ZmMs20

The F_2_ population derived from the cross of *ms20* mutant × Chang7-2 was used for gene mapping of the *ms20* locus. Primary mapping was performed by using a maize 6K SNP chip and two genomic DNA bulked pools from ten fertile and sterile F_2_ individuals, respectively. Chromosome 1 was first selected as the candidate chromosome of the *ms20* locus, since more than 80-Mb successive regions on its long arm have polymorphic ratios of > 40% (Figure 2A). Therefore, the *ms20* locus was primary-mapped to the long arm of chromosome 1. Then, 96 F_2_ individuals were randomly selected and used for gene mapping to verify the above result based on the SNP chip. Consequently, the *ms20* locus was located in a 4.8-cM region between markers bnlg2295 and P2-04 (Figure 2B, Appendix A). By using a large population, including 540 sterile F_2_ individuals, *ms20* locus was further narrowed down to a 190-kb interval between markers P9-08 and P9-12 (Figure 2C). Seven candidate genes were predicted in this region according to the B73 reference genome (AGPv4) (Figure 2D), including *Zm0001d029683*, which is the reported GMS gene, *IPE1* [30].

Among the seven candidate genes, *Zm00001d029685* was annotated as a provisional gene, and its expression was not detected in the following RNA-sequencing (RNA-seq) analysis. Expression patterns of the other six genes were analyzed based on RNA-seq data of maize anther during eight developmental stages (S5 to S11) (Figure 2E). *Zm00001d029680*, *Zm00001d029684*, and *Zm00001d029686* keep a persistent expression at a relatively low level, and expression of *Zm00001d029681* and *Zm00001d029685* was almost undetectable in anther. *Zm0001d029683* displayed high expression at S8b and S9 (Figure 2E). In addition, genomic DNA sequencing analysis of *Zm0001d029683* between WT and *ms20* mutant revealed that an 891-bp fragment was inserted at the -68 bp site of the 5′ UTR in *ms20* mutant and the coding region of *Zm0001d029683* was identical to that of WT (Figure 2F), while no DNA sequence difference of the other expressed candidate genes was observed between WT and *ms20* mutant. Therefore, all the results obtained above indicated that *Zm0001d029683* is the target gene and designated as *ZmMs20*, which is a new allele of *IPE1*.

### 2.4. Phylogenetic Evolution of ZmMs20

*ZmMs20* contains a 1743-bp open reading frame encoding 580 amino acids, and was predicted to encode a GMC oxidoreductase [30]. To illustrate the evolution relationship of ZmMs20, phylogenetic analysis was performed based on 19 orthologs in 11 plant species. The consequent neighbor-joining tree was clustered into two groups. The group I members were from monocots, and the group II members were from dicots (Figure 3A). Interestingly, only one ortholog was found in each of the selected monocot species, except wheat, which is hexaploidy and has 3 orthologs of *ZmMs20*. However, multiple orthologs were commonly presented in each of the selected dicot species except *Arabidopsis* (Figure 3A). This suggests that there may be different evolution pathways of *ZmMs20* and its orthologs between monocots and dicots.

GMC oxidoreductase is a small gene family in plants. 8, 7, and 6 members were found in *Arabidopsis*, rice, and maize, respectively. Multiple sequence alignment and phylogenetic analysis were performed by using the total 21 GMC oxidoreductases in the three plants. Consequently, these GMC oxidoreductases were divided into 3 clades (Figure 3B). Clade I includes 7 members from rice and maize, and they are paired with each other for Os03t0118700. According to the previous reports, maize GMC family genes in clade I are specifically expressed in the tassel at the meiotic stage [32]. *Arabidopsis* At1g12570, an ortholog of ZmMs20, is the only member of clade II. In clade III, 13 members can be further divided into two subclades. Subclade III-1 includes 3 members from *Arabidopsis*, and subclade III-2 contains 10 members. Among the subclade III-2 members, MINI1/ONI3 (Os09t0363900) and ACE/HTH (At1g72970) were found to be involved in biosynthesis of long-chain fatty acids to prevent inappropriate fusions between neighboring floral organs [24,26].

Since all the reported GMC oxidoreductases in *Arabidopsis* and rice were involved in floral development, we analyzed expression patterns of the 6 GMC family genes in maize anther by using RNA-seq data during 8 developmental stages (Figure 3C, Appendix A). Five of the 6 genes displayed high expression at certain anther stages, while the expression level of *Zm00001d032284* was low, similar to its rice ortholog *Os08t0401500*, whose transcript was not detectable in rice [25]. *Zm00001d002613*, *Zm0001d017598*, and *Zm00001d020238* displayed similar expression patterns to *ZmMs20* with the expression peak at S9, while the expression of *Zm00001d036701* appeared at S6, raised a plateau at S7 to S8b, and disappeared at S9 (Figure 3D). These results suggest that most of maize GMC oxidoreductases are perhaps involved in maize anther development.

In addition, microsynteny assay was performed by using flanking genes of *ZmMs20* and its orthologs from six gramineous plants, including *Brachypodium distachyon* (slender falsebrome), *Hordeum vulgare* (barley), rice, *Sorghum bicolor* (sorghum), *Setaria italica* (millet), and maize (Appendix A), and showed greatly significant synteny between *ZmMs20* and its orthologs among the six grasses (Figure 4). This result indicated that evolution of chromosomal regions harboring *ZmMs20* and its orthologs is relatively conserved in these gramineous plants.

### 2.5. The Rigorous Stage-specificity Expression Patterns of GMS Genes in Maize

Expression patterns of *ZmMs20* were evaluated by using RNA-seq data of WT anthers from S5 to S11. The transcript level of *ZmMs20* appeared at S8a and peaked at 8b and S9 (Figure 5A,B4). To find out the sequential relationship of gene expression, we analyzed expression patterns of the 17 cloned GMS genes in maize and clustered these genes into four clades (Figure 5, Appendix A). Clade I includes *OCL4* and *Ms22*/*MSCA1* [33,34], whose expression level peaked at S5 and became almost undetectable after S6 (Figure 5A,B1). Clade II contains *MAC1*, *Ms32*, and *Ms33*, whose expression started at S5 and peaked at S6, and then *MAC1* and *Ms32* [35,36] were expressed in relatively low levels until S11, but expression of *Ms33* [15] almost disappeared after S7 (Figure 5A,B2). Clade III is composed of *Ms23*, *Ms9*, *Ms8*, *ZmMs30*, and *ms44* [17,37,38,39,40], which shared a similar expression plateau from S7 to S8b (Figure 5A,B3). Clade IV contains *Ms45*, *Ms7*, *Ms20*/*IPE1*, *Ms26*, *APV1*, *IG1*, and *Ms6021* [7,10,13,20,30,31,41], whose expression peaked at S9 and then decreased significantly (Figure 5A,B4). All the GMS genes displayed rigorous stage-specificity expression patterns during maize anther development, suggesting that different GMS genes may play important roles in regulating anther and pollen development and determining male fertility at their corresponding functioning stages in maize.

### 2.6. Two Co-segregating Functional Markers Developed for Creating ms20 Male-sterility Lines by MAS Strategy under Different Genetic Backgrounds

GMS lines have exhibited great value for hybrid breeding and seed production. Development of GMS lines can be accelerated by using co-segregating functional markers for MAS. Based on the information of 891-bp insertion in *ms20* mutant, two makers, i.e., ms20-IN1 and ms20-IN2, were designed for MAS. The forward (1F) and reverse (1R) primers of ms20-IN1 were located upstream and downstream of 891-bp insertion, respectively (Figure 6A,B). The ms20-IN2 marker shared the forward primer (1F) with ms20-IN1, while its reverse primer (2R) was located in the 891-bp insertion (Figure 6A,B). Based on this design, 204-bp and 1095-bp PCR products should be amplified by using a 1F/1R primer pair in homologous WT and *ms20* mutant, respectively, and 487-bp PCR product should be amplified by using a 1F/2R primer pair in *ms20* mutant, but no PCR product could be detected in homologous WT (Figure 6B). The accuracy of two functional markers was further verified by using a PCR amplification test in three genotypic materials (*ZmMs20*/*ZmMs20*, *ZmMs20*/*ms20*, and *ms20*/*ms20*) as templates (Figure 6C). In addition, the co-segregating markers could differentiate genotypes of *ZmMs20* and *ms20* in both F_2_ and BC_1_F_1_ populations (Figure 6D,E). These results indicate that ms20-IN1 and ms20-IN2 markers can be used for MAS breeding to create *ms20* lines under different genetic backgrounds. To date, *ms20* mutation gene has been introduced into 353 different maize inbred lines by using these two function markers and the MAS strategy in our lab, and the obtained 353 F_2_ individuals with genotypes of *ms20/ms20* will be used to verify the male-sterility stability caused by *ms20* mutation under different genetic backgrounds, based on the method as described in An et al., 2019 [17].

## 3. Discussion

The *ms20*, a complete male-sterility mutant, displayed smaller and wilted anther, a glossy anther outer surface, disappearance of Ubisch bodies, and aborted pollen grains (Figure 1). Its fertility restorer gene, *ZmMs20*, is a new allele of *IPE1* [30] encoding a GMC oxidoreductase specifically expressed in maize anthers (Figure 2). Loss-function of *IPE1* leads to similar defective phenotypes as *ms20* mutant, for instance, absence of the typical Spaghetti-like anther cutin pattern, disappearance of Ubisch bodies, and arrested pollen exine accumulation [30]. Anther cuticle and pollen exine were considered as two important physical barriers protecting anther and pollen from various biotic and abiotic stresses [17], and their successful formation were found to be vital for anther and pollen development and male fertility.

The phylogenetic and microsynteny analyses revealed that *ZmMs20* and its orthologs were conserved in gramineous plants (Figure 3 and Figure 4). The identity of the amino acid was more than 88.8% among ZmMs20 and its orthologs. These results were consistent with the previous reports that the GMC domain was highly conserved and variations were primarily concentrated within the first 40 amino acids, which may contain a signal peptide [27]. *OsNP1*, an ortholog of *ZmMs20*, has been reported in rice [28,29]. Its mutant displays similar defective phenotypes to *ms20* mutant, such as undeveloped anther cuticle, disappearance of Ubisch bodies, and immature pollen exine. This implied conserved function between *ZmMs20* and *OsNP1*. *OsNP1* was predicted to take part in polymerization and assembly of sporopollenin and Ubisch bodies [29], which is essential for male reproduction processes. Therefore, we can conclude that orthologs of *ZmMs20* in sorghum, millet, and wheat may also play similar roles in regulating anther and pollen development and male fertility. Knock-out of these orthologs possibly creates male-sterility genetic materials, which are useful for cross-breeding and hybrid seed production in crops.

*ZmMs20* encodes a GMC oxidoreductase, a small gene family in plants. Phylogenetic analysis results indicated that GMC family genes might undergo different evolution pathways between monocots and dicots. Gene duplication had perhaps taken place in dicots to produce more GMC copies or members in dicots’ genome (Figure 3). In addition, mutations of *ZmMs20* orthologs result in different defects in *Arabidopsis* and rice, likely due to their functional differentiation. For instance, loss-function of *ACE*/*HTH*, the first GMC gene reported in *Arabidopsis*, leads to fusions of floral organs and genetic background-dependent male sterility due to the disruption of the membrane structure [24]. However, the early organ fusion of *oni3*/*mini1* mutant is mainly observed in rice vegetative organs, and results in severe defects, such as seedling lethality [25,26]. *Arabidopsis At1g12570*, an ortholog of *ZmMs20* and *OsNP1*, is the key regulator of cutin biosynthesis [42]. The T-DNA insertion mutant of *At1g12570* (SALK-085330C) exhibits normal anther cuticle development and pollen grain extrusion as WT, even though a quantity of smaller pollen grains is observed in the mutant and the surface of some pollen grains with normal sizes is smooth [30]. The mutant phenotypes are greatly different from those of *ms20* (Figure 1) and *np1* [29]. Taken together, it can be concluded that GMC-domain genes in dicots may retain partial functions for regulating anther and pollen development when compared to their corresponding orthologs in monocots, or the GMC genes are functionally redundant in dicots for gene duplication.

In maize, six GMC oxidoreductases were found according to the B73 reference genome (AGPv4). Apart for *Zm00001d032284*, whose ortholog (*Os08t0401500*) expression was undetectable in rice anther, all five other GMC genes displayed stage-specific expression patterns based on the RNA-seq analysis (Figure 3C,D). This indicates that the GMC family genes mainly contribute to anther development in maize, and thus systematic research on this gene family will provide clues to uncover how GMC family genes take part in the genetic regulation networks and bio-chemical pathways associated with anther development in maize.

All the maize GMS genes reported previously belong to either transcription factor or aliphatic enzymes. These GMS genes showed rigorous stage-specific expression patterns during anther development (Figure 5). Transcripts of *ZmMs20* and the clade-IV genes encoding aliphatic enzymes were found to appear after initiation of meiosis, peak at S9, and then decline radically (Figure 5B4). Therefore, exploring regulators or interaction partners of *ZmMs20* could contribute to deeply understanding their roles in regulating anther development. In addition, Figure 5 clearly indicates that there are certain aliphatic enzyme genes controlling anther development at each stage from S5 to S9, since the expression peak of these genes appears at S5 to S9 separately. This implies that multiple aliphatic enzymes function in sequence on anther development. The aliphatic metabolisms in anther tapetum can produce and provide precursors for anther cuticle and pollen exine formation [43], which are essential for male fertility. However, the metabolism pathways are very sophisticated and the related bio-chemical mechanism is still unclear. Therefore, deciphering the in vivo substrates of these aliphatic enzymes and exploring the functional interaction relationship among these genes would greatly benefit our understanding of bio-chemical processes during anther development. *ZmMs20*/*IPE1* was predicted to act downstream of *ZmMs26*, and could convert the ω-hydroxy C16/C18 fatty acids into C16/C18 diacids. C16:0 and C18:2 diacids were significantly reduced in the *ipe1* mutant anther [30], while C16 and C18 diacids only account for less than 1% of the total amount of cutin. How the trace of cutin component plays a vital role for anther development is still unknown. Therefore, more and deeper exploration needs to be performed in the future work to evaluate the exact substrates and catalytic activities of ZmMs20 and other aliphatic enzymes.

Finally, GMS genes and their corresponding mutants are valuable for hybrid breeding and seed production in crops. Most recently, the multi-control sterility (MCS) system has been developed, appraised, and updated by using *ZmMs7*, *ZmMs33*, and *ZmMs30* and their male-sterility mutants in our laboratory [16,17,20]. This provides an opportunity to tactfully settle some matters and effectively utilize GMS genes for hybrid vigor utilization and hybrid seed production. Hence, the further exploration of new GMS genes is of great significance. For breeding application of *ZmMs20* gene and *ms20* mutant, two functional markers were developed to facilitate creating more *ms20* male-sterility lines under wide genetic backgrounds based on MAS strategy (Figure 6).

In summary, this work contributes to both mechanism deciphering of anther development and breeding application of the GMS gene and its mutant for creating more male-sterility lines, which can be used for hybrid vigor utilization and seed production in maize.

## 4. Materials and Methods

### 4.1. Plant Materials, Growth Condition, and Phenotypic Characterization

The *ms20* (No. 928C) mutant seeds were originally obtained from the Maize Genetics Operation Stock Center (http://maizecoop.cropsci.uiuc.edu). All the plant materials were grown in the experiments stations of University of Science and Technology Beijing (USTB) in Beijing and Hainan Province. Tassels were photographed with an EOS 7000 digital camera (Canon, Tokyo, Japan). Anthers and pollen grains stained with 1% I2-KI solution were photographed with a SZX2-ILLB stereomicroscope (Olympus, Tokyo, Japan) and BX-53F microscope (Olympus, Tokyo, Japan), respectively.

### 4.2. SEM Analysis

For SEM observation, the fresh anthers of both WT and *ms20* mutant were immersed in FAA solution (Coolabor, Beijing, China) overnight for fixation. The anthers were then dehydrated in an ethanol series (50%, 75%, 95%, and 100%) for 15 min each step. After critical-point drying, the samples were coated with palladium gold in an ion sputter (JFC, Osaka, Japan) and observed under a SEM (HITACHI S, Tokyo, Japan).

### 4.3. Genetic Analysis and Map-Based Cloning of ZmMs20

F_2_ segregating population was derived from the cross of *ms20* × Chang7-2. Genetic analysis was performed by calculating the segregating ratio of fertile to sterile F_2_ individuals. Genomic DNA was extracted from maize leaves using a modified method [44].

To map *ms20* locus on chromosomes, ten F_2_ individuals of each phenotype (male-fertility and sterility plants) were randomly selected for constructing a fertility bulked DNA pool (*ZmMs20*) and male-sterility bulked DNA pool (*ms20*). A SNP polymorphism analysis was performed by using maize 6K SNP chip (Compass Biotechnology, Beijing, China) assay, and genotyping was performed according to the manufacturer’s recommendations using the Illumina iScan System (Illumina, San Diego, CA, USA). The polymorphic ratios between WT and *ms20* bulked pools in the 10-cM region along the chromosome was calculated, and the successive region with a high polymorphic ratio was considered as the candidate flanked region of *ZmMs20*.

For gene mapping of *ms20* locus, 96 F_2_ individuals were screened with SSR markers in the candidate-linked region. Genotyping data of these markers and F_2_ individual phenotypes were analyzed by using the Joinmap 4.0 program. Subsequently, fine mapping of *ms20* locus was carried out by using 540 F_2_ individuals with male-sterility phenotype and the developed SSR markers based on the B73 reference genome (AGPv4). Finally, the *ms20* locus was mapped to a 190-kb region on chromosome 1. Within this region, *Zm00001d029683* was considered as the target gene of *ZmMs20*. It was previously reported as *IPE1* [30].

### 4.4. Sequence Alignment and Phylogenetic Analysis

Multiple sequence alignment of maize ZmMs20 with its corresponding orthologs in other plants were performed in the CLUSTALX program [45]. The phylogenic tree was reconstructed in the Molecular Evolutionary Genetics Analysis (MEGA6) program using the maximum likelihood method [46]. Support values were estimated by 1000 times of bootstrap replicates.

### 4.5. Expression Analysis and Clustering of the Maize Cloned GMS Genes Using RNA-seq

For RNA-seq analysis, total RNAs of maize WT anthers from S5 to S11 were extracted using RNeasy Plant Mini Kit (Qiagen, Dusseldorf, Germany). Each sample was used to create libraries that were deep-sequenced using the Illumina HiSeq 2500 System (Illumina, San Diego, CA, USA) to generate 150-bp, paired-end reads. Reads were trimmed based on quality scores (*p* ≥ 15) and adapter sequences were removed. Reads were mapped to the maize reference genome (AGPv4) using TopHat2.0 with default parameters [47]. Aligned reads were counted with Rsubread and then quantified and normalized with edgeR [48,49]. Normalized expression is shown in RPKM (read per kb per million mapped reads). Pheatmap program was used to generate heat maps of expression levels of the 17 cloned GMS genes and further perform clustering and classification.

### 4.6. Microsynteny Analysis of ZmMs20

For microsynteny analysis, neighboring genes of *ZmMs20* were identified in the maize reference genome (AGPv4), and then the flanking genes of *ZmMs20* orthologs in sorghum, millet, rice, barley, and slender falsebrome were downloaded from Phytozome V12 (https://phytozome.jgi.doe.gov/pz/ portal.html) (Appendix A). Microsynteny of *ZmMs20* was determined by multiple sequence alignments of the flanking genes surrounding *ZmMs20* or its orthologs among the six grasses.

### 4.7. Development of the Co-segregating Markers of ZmMs20/ms20 for MAS Breeding

Based on the sequence differences of *ZmMs20* between WT and *ms20* mutant, two co- segregating markers, ms20-IN1 and ms20-IN2, were designed and tested by PCR amplifications with gel electrophoresis. The two markers could be used for MAS breeding of the *ms20* locus to introduce it into maize of different genetic backgrounds. All the PCR primers described above are listed in Appendix A.

## Figures and Tables

**Figure 1 ijms-20-01411-f001:**
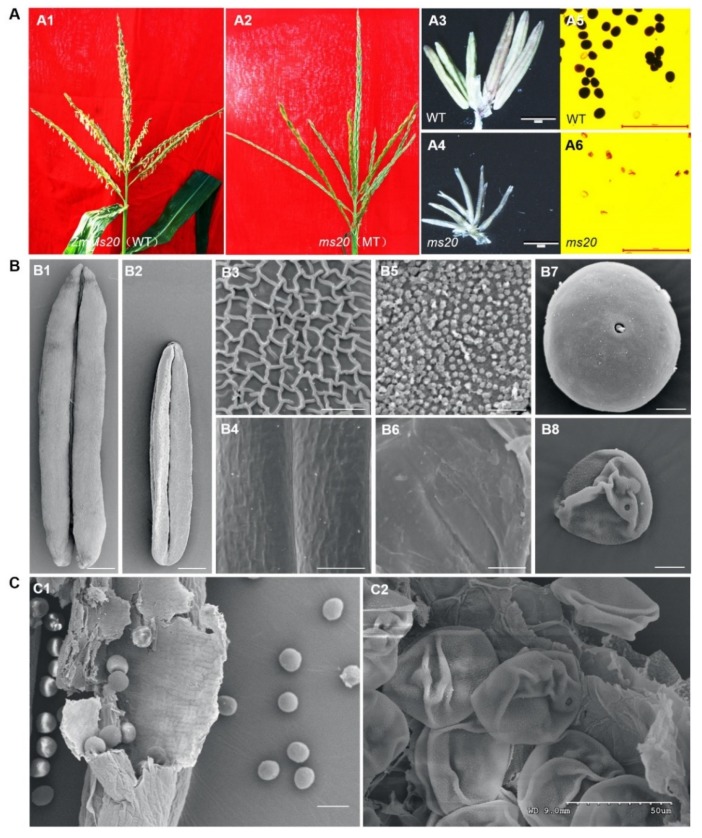
Phenotypic comparison of maize tassels, anthers, and pollen grains between wild type (WT) and *ms20* mutant at anthesis stage. (**A**) Tassels (**A1** and **A2**), spikelet (**A3** and **A4**), and pollen grains stained with I_2_-KI solution (**A5** and **A6**) of WT and *ms20*. Bars = 2.5 mm in **A3** and **A4**, 500 μm in **A5** and **A6**. **(B**) SEM analysis of WT and *ms20* anthers and pollen grains at anthesis stage. (**B1**,**B2**) The *ms20* anther (**B2**) is smaller and wilted than that of WT (**B1**). (**B3**,**B4**) The outer surface of *ms20* anther (**B4**) is glossy compared with that of WT (**B3**). (**B5**,**B6**) Ubisch bodies (**B5**) were observed on the inner surface of WT anther but none on the *ms20* anther (**B6**). (**B7**,**B8**) The WT pollen grain is round-shape (**B7**), while *ms20* pollen is wizened in the locule (**B8**). Bars = 500 μm in **B1** and **B2**, 5 μm in **B3** and **B4**, 2 μm in **B5** and **B6**, and 10 μm in **B7** and **B8**. (**C**) SEM analysis of WT and *ms20* pollens at anthesis stage. For **C1** and **C2**, WT pollens (**C1**) are spread out, but *ms20* pollens (**C2**) are stuck on the inner surface of anther locule. Bars = 100 μm in **C1**, and 50 μm in **C2**.

**Figure 2 ijms-20-01411-f002:**
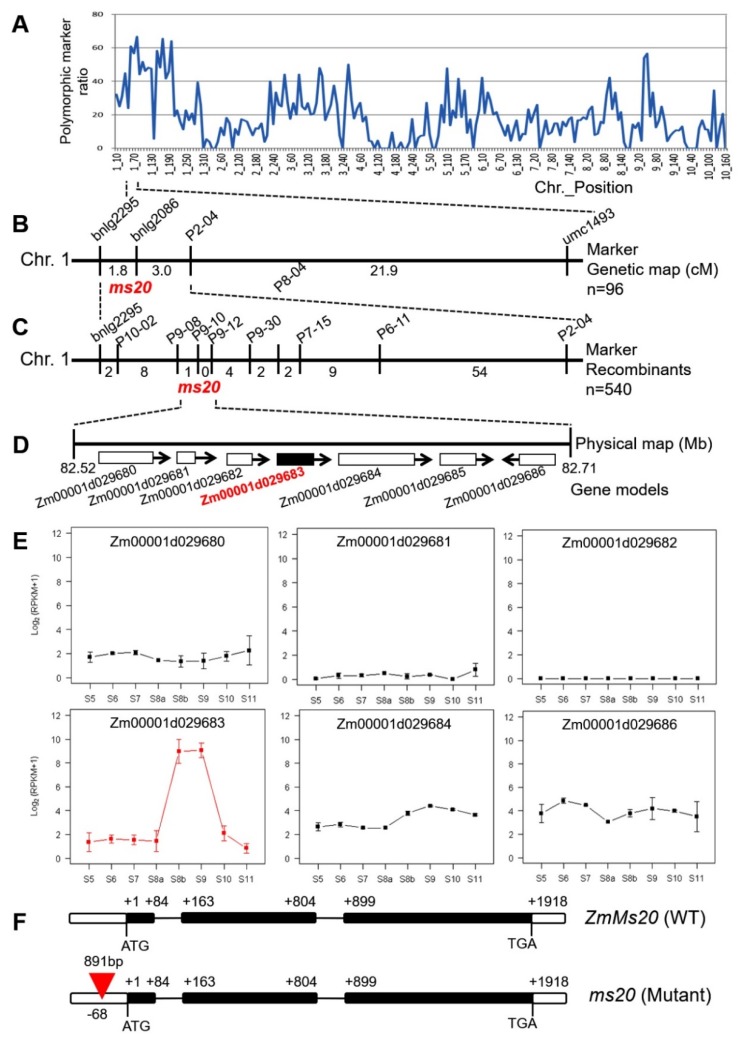
Map-based cloning of maize *ms20* gene, expression patterns of six candidate genes, and gene structure of *ZmMs20*. (**A**) Polymorphic marker ratio of the sterile plant DNA pool (*ms20*/*ms20*) versus the fertile plant pool (*ZmMs20*/*ZmMs20* and *ZmMs20*/*ms20*). (**B**) Primary mapping of *ms20* locus. The *ms20* locus was primarily mapped to the long arm of maize chromosome 1 between Simple Sequence Repeat (SSR) markers bnlg2295 and P2-04; n, the number of F_2_ plants used for gene mapping, including 48 male-sterility and 48 male-fertility F_2_ individuals. (**C**) Fine mapping of *ms20* locus. The *ms20* locus was narrowed down to a 190-kb interval between SSR markers P9-08 and P9-12; n, the number of male-sterility F_2_ individuals used for fine mapping. (**D**) The seven candidate gene models in the 190-kb interval. Among them, *Zm00001d029683* was the *IPE1* gene reported previously by Chen et al., 2017 [30]. (**E**) Expression patterns of six of the seven candidate genes in the 190-kb interval based on RNA sequencing data during 8 anther developmental stages (S5 to S11). *Zm00001d029685* was annotated as a provisional gene and its expression was not detectable in the RNA sequencing data, whereas *Zm00001d029683* was found to be high expression at both stages 8b and 9. (**F**) Gene structure of *ZmMs20* and DNA sequence mutation of *ms20*. *ZmMs20* consists of three exons and two introns. The +1 indicates the starting nucleotide site of translation, and the stop codon (TGA) is +1918. Black boxes indicate exons, and intervening lines indicate introns. An 891-bp insertion at the -68 bp site of the 5′ UTR was found in the *ms20* mutant, and shown in a red triangle.

**Figure 3 ijms-20-01411-f003:**
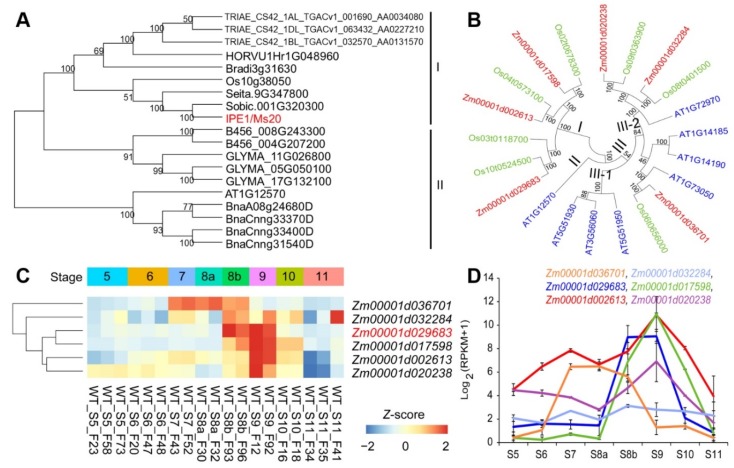
Phylogenetic analysis of ZmMs20 and its orthologs in different species and expression pattern analysis of *ZmMs20* and its homologies in maize. (**A**) Phylogenetic analysis of ZmMs20 and its orthologs in different species. The analysis involved 19 amino acid sequences from *Arabidopsis thaliana* (At), *Brassica napus* (Bna, rape), *Bradchypodium distachyon* (Bradi, slender falsebrome), *Hordeum vulgare* (HORVU, barley), *Gossypium raimondii* (B, cotton), *Glycine max* (GLYMA, soybean), *Oryza sativa* (Os, rice), *Setaria italica* (Seita, millet), *Sorghum bicolor* (Sobic, sorghum), *Triticum aestivum* (TRIAE, wheat), and *Zea mays* (*IPE1*/*Ms20*, maize). The numbers on the branches represent the bootstrap values of the phylogenetic tree. The 19 orthologs can be divided into two groups. Group I comprises nine proteins from monocots, and group II includes ten proteins from dicots. (**B**) Phylogenetic tree of GMC family members in *Arabidopsis*, rice, and maize. A neighbor-joining tree showed the evolutionary relationships among GMC members in *Arabidopsis* (At), rice (Os), and maize (Zm). The GMC family members are named according to maize GDB or Phytozome accession number. The numbers under the branches represent the bootstrap values. (**C**) Hierachical clustering of six GMC family genes in maize based on RNA-seq data of anthers during 8 anther developmental stages (S5 to S11). (**D**) Expression patterns of *ZmMs20* and its five homologies in maize.

**Figure 4 ijms-20-01411-f004:**
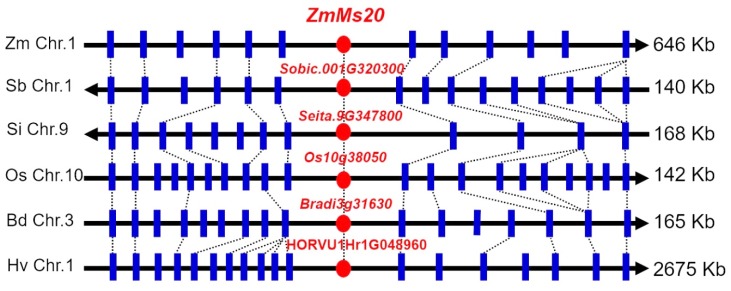
Microsynteny comparison of maize *ZmMs20* locus with its orthologs in five other monocot species. Microsynteny comparison of maize *ZmMs20* locus with its ortholog counterparts from *Brachypodium distachyon* (Bd), *Hordeum vulgare* (Hv), *Oryza Sativa* (Os), *Sorghum bicolor* (Sb), and *Setaria italica* (Si). The red solid circles indicate *ZmMs20* and the orthologous genes, and the blue solid vertical lines indicate the genes flanking *ZmMs20* or the orthologous genes. The arrows indicate direction of the chromosomes (Chr.), and each gene in the flanking region was given a serial number (Refer to Appendix A). The spanning region length between two loci was shown without equal proportion, and total length (kilobase, kb) of the ortholog region was shown on the right.

**Figure 5 ijms-20-01411-f005:**
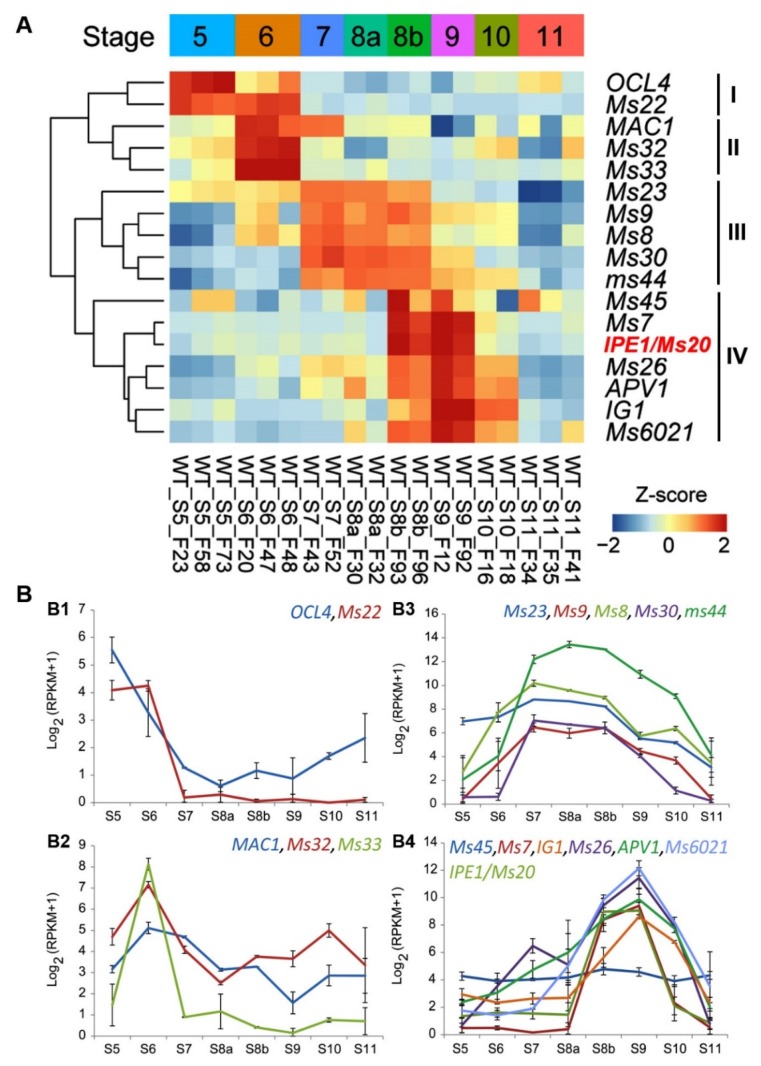
Expression patterns and classification of 17 GMS genes cloned in maize. (**A**) Hierachical clustering of the 17 cloned GMS genes based on RNA-seq data during eight anther developmental stages (S5 to S11). These GMS genes are clustered into four clades in chronological order of gene expression. Among them, *ZmMs20*/*IPE1* is clustered into the clade IV. (**B**) Expression patterns of the GMS genes in clade I (**B1**), clade II (**B2**), clade III (**B3**), and clade IV (**B4**).

**Figure 6 ijms-20-01411-f006:**
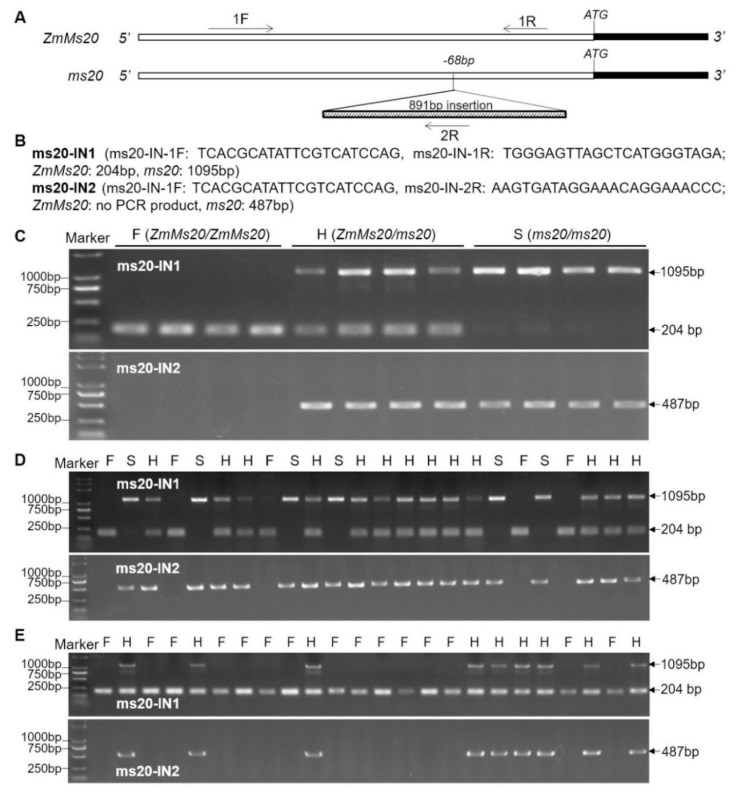
The functional markers of *ZmMs20*/*ms20* and MAS of *ms20* mutant locus to introduce it into different genetic backgrounds. (**A**) A schematic representation of functional marker locations. The 1F represents the common forward primer of ms20-IN1 and ms20-IN2 markers. The 1R represents the reverse primer of ms20-IN1 marker, and 2R represents the reverse primer of ms20-IN2 marker. (**B**) The primers’ sequence and PCR production sizes of ms20-IN1 and ms20-IN2 markers. (**C**) Detection and verification of ms20-IN1 and ms20-IN2 markers. For ms20-IN1 marker, the larger bands with 1095-bp length indicate *ms20* loci (sterility, S), the smaller bands with 204-bp length mean *ZmMs20* loci (fertility, F), and H means heterozygous genotype of *ZmMs20*/*ms20*. For ms20-IN2 marker, the bands with 487-bp length indicate *ms20* loci (sterility, S), and the no PCR production means *ZmMs20* loci (fertility, F). (**D**) Genotyping of F_2_ population derived from the cross and self-pollination of *ms20* mutant and a maize inbred line Zheng 58 using ms20-IN1 and ms20-IN2 markers. (**E**) Genotyping of BC_1_F_1_ population derived from the cross and backcross of *ms20*/Zheng 58//Zheng 58 using *ms20*-IN1 and ms20-IN2 markers.

**Table 1 ijms-20-01411-t001:** The ratio of fertile to sterile plants in the F_2_ population derived from the cross of *ms20*×Chang7-2.

F_2_ Population Combination	Total Plants	Fertile Plants (F)	Sterile Plants (S)	F/S Ratio	χ^2^	*P*	Significant Test *p* > 0.05
*ms20*×Chang7-2	1365	1065	300	3.55:1	3.47	0.06	ns

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
