# Peer review of "Map-Based Cloning, Phylogenetic, and Microsynteny Analyses of ZmMs20 Gene Regulating Male Fertility in Maize"

_ijms, 2019, doi:10.3390/ijms20061411_

Round 1
Reviewer 1 Report
I have read the manuscript and have the following comments.
In the Abstract, please state the reasons for undertaking this research, the objectives, the innovations, and the usefulness of your results for the agronomists and breeders.
The Introduction should be restructured. The introduction should give a clear picture of the research study, along with its objectives and significance. Please state clearly the objectives of this research along with the innovations.
I find that the title does not reflect accurately the research. From what I understood, there was no breeding application of ZmMs20 since the authors did not use the gene to make selections in a breeding program. They developed and tested two functional markers for creating ms20 male-sterility lines, but no practical selection or hybrid seed production was performed in this study.
The authors report a lot of information on phylogenetic and microsynteny analysis, yet this is not reflected in the title. A more appropriate title would be: “Map-based cloning, phylogenetic and microsynteny analyses of ZmMs20 gene regulating male fertility in maize.”
I suggest that the authors revise their manuscript according to this title and the data they present.
Authors report that the male-sterility mutant ms20 was obtained from the Maize Genetics Stock Center. Why did the authors choose to work with ms20 versus other possible mutants? Was it chosen randomly? Please justify your selection of ms20 and state the reasons that you decided to work with this particular genetic mutant.
Pages 4 and 5: Why did you use the cross of ms20 x Chang7-2 for the genetic analysis and gene mapping? Why did you select 96 F2 individuals and 540 sterile F2 individuals? You need to justify your choices.
Figures and References are in good shape. Figure 1 is very nice.
Please revise your manuscript accordingly, give a clear picture of the objectives, state the innovations (I could not find any novel ideas) and the significance of the results. I suggest that you put more emphasis on the phylogenetic and microsynteny results and remove statements about the breeding application, unless you have more data to present on the breeding application.
Author Response
Responses to Reviewer-1’s Comments
Thank you for your conscientious work and pertinent comments on our manuscript. All the comments were responded point-by-point as below.
Point 1: In the Abstract, please state the reasons for undertaking this research, the objectives, the innovations, and the usefulness of your results for the agronomists and breeders.
Response 1:
(1) The reasons for undertaking this research, the objectives:
Genic male sterility (GMS) mutant is a useful germplasm resource for hybrid vigor utilization and hybrid seed production. The identification and characterization of GMS genes in maize have deepened our understanding of the molecular mechanisms controlling anther and pollen development, and enabled the development and efficient use of many biotechnology-based male-sterility (BMS) systems for crop hybrid breeding. To obtain more maize GMS genes, we performed this study.
(2) the innovations, and the usefulness of your results for the agronomists and breeders:
a) We reported a complete GMS mutant (ms20), which displays abnormal anther cuticle and pollen development. Its fertility restorer gene ZmMs20 was isolated by map-based cloning strategy and found to be a new allele of IPE1 encoding a glucose methanol choline (GMC) oxidoreductase involved in lipid metabolism in anther.
b) Microsynteny analysis of ZmMs20 was performed for the first time, which revealed that ZmMs20 and its orthologs were conserved among gramineous species. This provides clues for the creation of GMS material using ZmMs20 homologous gene for other crops.
c) Six GMC genes were found in the maize reference genome, and were classified into different clades according to the phylogenetic analysis. The expression pattern analysis of all the six GMC genes was performed based on RNA-seq data of anthers during 8 anther developmental stages (S5 to S11). Five out of six GMC genes displayed stage specific expression in anther, which implied that most GMC genes participate in maize anther development. Additionally, expression pattern of 17 cloned maize GMS genes was analysed based on the same RNA-seq data. ZmMs20 showed similar expression pattern with Ms7, Ms26, Ms6021, APV1, and IG1 genes, which given some cues for deciphering their functional relationships in regulating male fertility.
d) For the application of male-sterility mutants in breeding, functional markers would facilitate the creation and selection of male-sterility lines under different genetic background based on marker-assisted selection (MAS) strategy. ms20 gene has been introduced into 353 different genetic background based on these two function markers, which would be used to verify the male-sterility stability caused by ms20 mutation in our lab.
Point 2: The Introduction should be restructured. The introduction should give a clear picture of the research study, along with its objectives and significance. Please state clearly the objectives of this research along with the innovations.
Response 2:
ms20 is a new allele mutation of ipe1, whose genetic and cytological characteristics have been reported in detail (Chen et al., 2017). In this study, we focused on the phylogenetic analysis of GMCs, expression pattern analyses of all GMC genes in maize and cloned maize GMS genes, and development of two functional markers of ms20 for creating more male-sterile lines under different genetic background in maize breeding program, which provided some new opinions as stated in the restructured introduction section in the revised manuscript. The objectives and innovations of this study were presented in the above Response 1.
Point 3: I find that the title does not reflect accurately the research. From what I understood, there was no breeding application of ZmMs20 since the authors did not use the gene to make selections in a breeding program. They developed and tested two functional markers for creating ms20 male-sterility lines, but no practical selection or hybrid seed production was performed in this study.
The authors report a lot of information on phylogenetic and microsynteny analysis, yet this is not reflected in the title. A more appropriate title would be: “Map-based cloning, phylogenetic and microsynteny analyses of ZmMs20 gene regulating male fertility in maize.” I suggest that the authors revise their manuscript according to this title and the data they present.
Response 3:
Thank you for your insightful suggestion on the modification of title. Phylogenetic analysis is an important result in this study. We have revised the title to “Map-based cloning, phylogenetic and microsynteny analyses of ZmMs20 gene regulating male fertility in maize”, which is more appropriate.
The test of male-sterility stability of ms20 mutation under different genetic backgrounds is in progress in our lab. The two functional markers developed in this study was used for genotyping of three gene types (ms20/ms20, ZmMs20/ms20 and ZmMs20/ ZmMs20) to select ZmMs20/ms20 for self-pollination and developing F2 segregating population in this test, but the data was not shown in this publication. We added this content in Lines 253-254 and Lines 269-272 in the revised version.
Point 4: Authors report that the male-sterility mutant ms20 was obtained from the Maize Genetics Stock Centre. Why did the authors choose to work with ms20 versus other possible mutants? Was it chosen randomly? Please justify your selection of ms20 and state the reasons that you decided to work with this particular genetic mutant.
Response 4:
Dozens of male-sterility mutants were obtained from Maize Genetics Stock Centre. Then seven complete male-sterility mutants including ms1, ms7, ms20, ms25, ms30, ms33 and ms34 were chosen to create F2 populations for map-based cloning. As a result, ms7, ms30, and ms33 have been cloned and published previously in our lab (Plant Biotechnology Journal, 2018, 16: 459-471; Molecular Plant, 2019, 12(3), https://doi.org/10.1016/j.molp.2019.01.011; Theoretical and Applied Genetics, 2018, 131:1363-1378), and the deciphering work on ms1, ms25 and ms34 is in progress. Therefore, the research results of ms20 mutant and its fertility restorer gene ZmMs20 are used for writing this manuscript.
Point 5: Pages 4 and 5: Why did you use the cross of ms20 x Chang7-2 for the genetic analysis and gene mapping? Why did you select 96 F2 individuals and 540 sterile F2 individuals? You need to justify your choices.
Response 5:
It is well known that any normal maize inbred line can be used as male parent to cross with the mutant ms20 for performing genetic analysis and constructing gene mapping population. Chang7-2 is an elite inbred line with robust tassels, so we chose it as the male parent to obtain more hybrid F1 seeds for developing a larger F2 mapping population easily.
In order to facilitate the experimental operation, 96 F2 individuals were selected for primary mapping of ms20 because the PCR plate contains 96 wells. The number of F2 individuals used for primary mapping, fine mapping and map-based cloning of one gene was not predetermined. It mainly relies on the location of the candidate gene on maize chromosome. The ms20 locus was narrowed down to a 190-kb interval using 540 F2 sterile individuals and molecular markers, and only 7 genes remained in this interval. Then DNA sequencing was performed to identify the candidate gene.
Point 6: Figures and References are in good shape. Figure 1 is very nice.
Please revise your manuscript accordingly, give a clear picture of the objectives, state the innovations (I could not find any novel ideas) and the significance of the results. I suggest that you put more emphasis on the phylogenetic and microsynteny results and remove statements about the breeding application, unless you have more data to present on the breeding application.
Response 6:
Thank you for your positive comments on Figures and References.
The objective, innovations and the significance of the results were stated in Response 1. The development of BMS based on GMS genes and corresponding male-sterility mutants benefits both hybrid vigor utilization and hybrid seed production. Therefore, the identification of new GMS genes and male-sterility mutants would provide genetic resources for BMS development. ms20 is a new allele of ipe1, and it displays complete male-sterility. This indicates that we provide new genetic material for BMS development in maize.
The male-sterility mutation used for BMS system must be stable under different genetic background. This work is in progress in our lab, and the two developed functional markers were used for genotyping to select ZmMs20/ms20 for self-pollination and developing F2 segregating population, and then F2 individuals with genotype of ms20/ms20 will be chosen to test whether or not complete male-sterility. So the statements about breeding application of ms20-cosegregating markers were remained in the revised manuscript. We added this content in Lines 253-254 and Lines 269-272 in the revised version.
Reviewer 2 Report
Overall comments:
This manuscript reported a male-sterility mutant ms20 and a new allele of the IPE1 gene which is associated with male sterility in maize. Two overall comments:
1) Results from this study looks very similar to Chen et al. 2017 who reported the IPE1 gene. It needs to be discussed what is the new contribution of this study compared to the previous study.
2) ZmMs20 was inconsistently referred to as gene and allele in this manuscript, which are two different concept and not interchangeable. According to the results (Page 6 line 165), the candidate gene Zm0001d029683 was the IPE1 gene as reported by Chen et al. (2017) and was renamed to be ZmMs20 in this manuscript. It needs to be clarified if ZmMs20 is a new allele of IPE1 or a new gene. If it is a new allele, please correct the use of “gene” throughout the manuscript. If it is a new gene, please prove what makes it different from IPE1?
Specific comments:
Page 3 Table 1: Consider widen the columns of F/S ratio, chi-square, and P, the numbers are difficult to read
Page 4 Line 129: “ms20 was mapped to the long arm of chromosome 1”, neither figure 2A nor the materials and methods explained well why the long arm of chromosome 1 was chosen. It is necessary to specify a threshold above which the polymorphic level was high.
Page 6 Line 167: “ZmMs20 contains….. and encodes GMC oxidoreductase”, no proof was provided in this study that ZmMs20 encodes GMC oxidoreductase. If this is based on Chen et al. (2017), it needs to be quoted here.
Page 7 Figure 3A. Please provide common names of the species along with their scientific names to be more readable.
Author Response
Responses to Reviewer-2’s Comments
Thank you for your careful review and insightful suggestions on our MS which would greatly improve this manuscript. We responded all the comments and suggestions point-by-point as below, and revised the manuscript according to your suggestions in the resubmitted version.
Point 1: Results from this study looks very similar to Chen et al. 2017 who reported the IPE1 gene. It needs to be discussed what is the new contribution of this study compared to the previous study.
Response 1:
ms20 is a new allele mutation of ipe1, whose genetic and cytological characteristics have been reported in detail (Chen et al., 2017). In this study, we focused on the phylogenetic analysis of GMCs, expression pattern analyses of all GMC genes in maize and cloned maize GMS genes, and development of two functional markers of ms20 for creating more male-sterile lines under different genetic background in maize breeding program, which provide some new opinions as below:
(1) Microsynteny analysis of ZmMs20 was performed for the first time, which revealed that ZmMs20 and its orthologs were conserved among gramineous species, which provide clues for the creation of GMS material using ZmMs20 homologous gene for other crops.
(2) Six GMC genes were found in the maize reference genome, and were classified into different clades according to the phylogenetic analysis result. The expression pattern analysis of GMCs was performed based on RNA-seq data of anthers during 8 anther developmental stages (S5 to S11). Five out of six GMCs displayed stage specific expression in anther, which implies that most GMC genes participate in maize anther development. Additionally, expression pattern of 17 cloned maize GMS genes was analysed based on the same RNA-seq data. ZmMs20 showed similar expression pattern with Ms7, Ms26, Ms6021, APV1, and IG1 genes, which given some cues for deciphering their functional relationships in regulating male fertility.
(3) For the application of male-sterility mutants in breeding, functional markers would facilitate the creation and selection of male-sterility lines under different genetic background based on marker-assisted selection (MAS) strategy. ms20 gene has been introduced into 353 different genetic background based on these two function markers, which would be used to verify the male-sterility stability caused by ms20 mutation in our lab.
These points were stated in the introduction and discussion section in the revised manuscript.
Point 2: ZmMs20 was inconsistently referred to as gene and allele in this manuscript, which are two different concept and not interchangeable. According to the results (Page 6 line 165), the candidate gene Zm0001d029683 was the IPE1 gene as reported by Chen et al. (2017) and was renamed to be ZmMs20 in this manuscript. It needs to be clarified if ZmMs20 is a new allele of IPE1 or a new gene. If it is a new allele, please correct the use of “gene” throughout the manuscript. If it is a new gene, please prove what makes it different from IPE1?
Response 2:
An 891-bp insertion at -68 bp site of Zm0001d029683 was found in the ms20 mutant, which is different from any ipe1 mutants reported by Chen et al. (2017). So ms20 mutation locus is a new allele mutation of ipe1 mutations reported in Chen et al. (2017). Moreover, the aborted pollen grains were observed in the ms20 mutant anther by I2-KI solution staining and the SEM assay, which was not in accordance with ipe1 mutant phenotype. We believed that the phenotypic difference likely results from sampling and investigating stages of maize anther. Therefore, ZmMs20 is a new allele gene of IPE1, and this was stated in the result. We named the new allele gene as ZmMs20 gene because it was cloned from ms20 male-sterile mutant.
Point 3: Page 3 Table 1: Consider widen the columns of F/S ratio, chi-square, and P, the numbers are difficult to read.
Response 3:
The columns width of Table 1 was resized to make sure numerical values separated from each other obviously.
Point 4: Page 4 Line 129: “ms20 was mapped to the long arm of chromosome 1”, neither figure 2A nor the materials and methods explained well why the long arm of chromosome 1 was chosen. It is necessary to specify a threshold above which the polymorphic level was high.
Response 4:
Thanks for your suggestion. We are sorry for not providing the mapping information in detail. In the revised manuscript, the details on chromosome selection and the threshold of the polymorphic level have been provided from Line 130 to Line 132.
Point 5: Page 6 Line 167: “ZmMs20 contains….. and encodes GMC oxidoreductase”, no proof was provided in this study that ZmMs20 encodes GMC oxidoreductase. If this is based on Chen et al. (2017), it needs to be quoted here.
Response 5:
This statement is based on Chen at al. (2017), and the literature was quoted here in the revised manuscript (Line 172).
Point 6: Page 7 Figure 3A. Please provide common names of the species along with their scientific names to be more readable.
Response 6:
For the aesthetic sense of figure and more readable, the common names of species were added along with their scientific names in the figure legend of Figure 3A.
Reviewer 3 Report
The authors have identified a complete male sterility mutant (“ms20”) and developed two functional markers for creating maize ms20male-sterility lines, which I think is interesting and useful in hybrid seed production in maize. Authors have done great job in explaining their results and discussion. However, I have provided few minor suggestions to consider before its publication.
Please clearly explain the rationale or gap of the study in the introduction section rather that reporting the results.
General comments applied throughout the manuscript
Add a comma after using three or more words
Make sure gene names are italicized consistently throughout the manuscript
Specific comments:
Line 32: Please do not repeat the keywords that are already present in the title such as Maize and ZmMs20
Line 50: “Up to date” should be replaced with “to date”
Line 51: Please write the scientific names of the crops, when used for the first time.
Author Response
Responses to Reviewer-3’s Comments
The authors have identified a complete male sterility mutant (“ms20”) and developed two functional markers for creating maize ms20 male-sterility lines, which I think is interesting and useful in hybrid seed production in maize. Authors have done great job in explaining their results and discussion. However, I have provided few minor suggestions to consider before its publication.
Response:
We are grateful to you for your careful review, active comments and insightful suggestions on this manuscript. We responded all the suggestions and refined the manuscript according to your suggestions in the resubmitted version.
Point 1: Please clearly explain the rationale or gap of the study in the introduction section rather that reporting the results.
Response 1:
ms20 is a new allele mutation of ipe1, whose genetic and cytological characteristics have been reported in detail (Chen et al., 2017). In this study, we focused on the phylogenetic analysis of GMCs, expression pattern analysis of GMCs and cloned GMS genes, and development of two functional markers of ms20 for creating more male-sterile lines in maize breeding program, which provide some new opinions as below:
(1) Microsynteny analysis of ZmMs20 was performed for the first time, which revealed that ZmMs20 and its orthologs were conserved among gramineous species, which provide clues for the creation of GMS material using ZmMs20 homologous gene for other crops.
(2) Six GMC genes were found in the maize reference genome, and were classified into different clades according to the phylogenetic analysis result. The expression pattern analysis of GMCs was performed based on RNA-seq data of anthers during 8 anther developmental stages (S5 to S11). Five out of six GMCs displayed stage specific expression in anther, which implies that most GMC genes participate in maize anther development. Additionally, expression pattern of 17 cloned GMS genes was analysed based on the same RNA-seq data. ZmMs20 showed similar expression pattern with Ms7, Ms26, Ms6021, APV1, and IG1 genes, which given some cues for deciphering their functional relationships in regulating male fertility.
(3) For the application of male-sterility mutants in breeding, functional markers would facilitate the creation and selection of male-sterility lines under different genetic background based on marker-assisted selection (MAS) strategy. ms20 gene has been introduced into different genetic background based on these two function markers, which would be used to verify the male-sterility stability caused by ms20 mutation in our lab.
These points were stated in the introduction and discussion section in the revised manuscript.
Point 2: Add a comma after using three or more words.
Response 2:
Thank you for your professional suggestion on punctuation usage. We checked throughout the manuscript and added a comma after using three or more words.
Point 3: Make sure gene names are italicized consistently throughout the manuscript.
Response 3:
The names of genes, proteins, and mutants were gone over and corrected throughout the manuscript.
Point 4: Line 32: Please do not repeat the keywords that are already present in the title such as Maize and ZmMs20.
Response 4:
In the keywords, Maize was replaced with Zea may, and ZmMs20 was replaced with “anther development” to describe the biological process ZmMs20 participates in (Lines 34-35).
Point 5: Line 50: “Up to date” should be replaced with “to date”.
Response 5:
“Up to date” was replaced with “To date” according to your suggestion in the updated version of manuscript (Line 51).
Point 6: Line 51: Please write the scientific names of the crops, when used for the first time.
Response 6:
Scientific names of crops were used when appeared for the first time and the common names were added along with their scientific names as annotation (as shown in Line 53).
Round 2
Reviewer 1 Report
I comment the authors for doing a great job replying to my comments. Some well-written sentences of their response are not included in the manuscript. I am happy with most of the corrections. I believe that the abstract can be further improved.
In the abstract, please clearly state the objectives of your study and the innovations for the breeders and the agronomists. Make sure you identify the objectives as 1 and 2, if there is more than one objective. What is the major innovation?
Other than that, the revised manuscript is in good shape.
Author Response
Responses to Reviewer-1’s Comments
Point 1: I comment the authors for doing a great job replying to my comments. Some well-written sentences of their response are not included in the manuscript. I am happy with most of the corrections. I believe that the abstract can be further improved.
In the abstract, please clearly state the objectives of your study and the innovations for the breeders and the agronomists. Make sure you identify the objectives as 1 and 2, if there is more than one objective. What is the major innovation?
Other than that, the revised manuscript is in good shape.
Response:
We are grateful for your positive comments on our responses and revised manuscript. In the revised manuscript of Round 2, we stated the objectives as 1) and 2) (Lines 20-21), which represent the theory and application significances of this study. Additionally, once the ms20 mutant displays stable male-sterility under wide genetic background, the ZmMs20 gene and ms20 mutant would be used to create maintainer line for hybrid seed production. This is the innovation for the breeders and agronomists.
The statements about objectives and innovations were briefly summarized in the revised abstract due to the length limitation. Here, we illustrated in detail as follows.
The objectives of the study:
1) Deepening the understanding of the molecular mechanisms controlling anther and pollen development. Anther is a good research model for metagenesis, which is a complicated biological process needing the cooperation of sporophyte and gametophyte. ms mutants provide valuable genetic materials for deciphering mechanism of anther development. Cytological observations on anther defects of mutants and molecular characterization on ms restorer genes provide clues for understanding the underlying mechanisms regulating anther and pollen development from different perspectives.
2) Providing ms restorer gene and ms mutant to develop the BMS system for hybrid breeding and seed production. ms mutants are valuable genetic resources for hybrid seed production. Compared to the cytoplasmic male sterility (CMS), the GMS line is controlled by a single recessive nuclear gene, and any normal maize germplasm can complement the male-sterility phenotype. However, the GMS lines can’t be propagated with a large scale by self-pollination. The BMS strategy based on GMS restorer gene and its corresponding ms mutant, such as MCS (constructed in our lab, Zhang et al, 2018) and SPT (Plant Biotechnology Journal, 2016, 14: 1046-1054), can settle the issue tactfully. The male sterility of ms mutant used in BMS system needs to be highly stable in various genetic backgrounds and environments. Additionally, the ms mutation must have no negative effect on any agronomic traits. Therefore, the male-sterility stability and potential effects of the ms mutation on many agronomic traits need to be assessed by combining the marker-assisted selection (MAS) approach and the method as described by An et al. (2019) in our lab. This study will provide a new ms mutant allele (ZmMs20) for developing BMS system.
The innovations of the study:
The innovation for the breeders and the agronomists was added in Lines 31-32, and elaborated in detail in the second point of objectives mentioned above, which is the major innovation for production practice in maize.
The revised Abstract as follows:
Abstract: Genic male sterility (GMS) mutant is a useful germplasm resource for both theory research and production practice. The identification and characterization of GMS genes, and assessment of male-sterility stability of GMS mutant under different genetic backgrounds in Zea may (maize) have 1) deepened our understanding of the molecular mechanisms controlling anther and pollen development, and 2) enabled the development and efficient use of many biotechnology-based male-sterility (BMS) systems for hybrid breeding. Here, we reported a complete GMS mutant (ms20), which displays abnormal anther cuticle and pollen development. Its fertility restorer gene ZmMs20 was found to be a new allele of IPE1 encoding a glucose methanol choline (GMC) oxidoreductase involved in lipid metabolism in anther. Phylogenetic and microsynteny analyses showed that ZmMs20 was conserved among gramineous species, which provide clues for creating GMS materials in other crops. Additionally, among the 17 maize cloned GMS genes, ZmMs20 was found to be similar to the expression patterns of Ms7, Ms26, Ms6021, APV1, and IG1 genes, which will give some cues for deciphering their functional relationships in regulating male fertility. Finally, two functional markers of ZmMs20/ms20 were developed and tested for creating maize ms20 male-sterility lines in 353 genetic backgrounds, and then an artificial maintainer line of ms20 GMS mutation would be created by using ZmMs20 gene, ms20 mutant and BMS system. This work will promote our understanding to functional mechanism of male fertility and facilitate molecular breeding of ms20 male-sterility lines for hybrid seed production in maize.